# Multi-Generation Ecosystem Selection of Rhizosphere Microbial Communities Associated with Plant Genotype and Biomass in *Arabidopsis thaliana*

**DOI:** 10.3390/microorganisms11122932

**Published:** 2023-12-06

**Authors:** Nachiket Shankar, Prateek Shetty, Tatiana C. Melo, Rick Kesseli

**Affiliations:** 1Department of Biology, University of Massachusetts, Boston, MA 02125, USArick.kesseli@umb.edu (R.K.); 2Institute of Plant Biology, Biological Research Centre, 6726 Szeged, Hungary; prateekshettys@gmail.com

**Keywords:** plant-associated microbiome, microbial diversity, soil health, sustainable agriculture

## Abstract

The role of the microbiome in shaping the host’s phenotype has emerged as a critical area of investigation, with implications in ecology, evolution, and host health. The complex and dynamic interactions involving plants and their diverse rhizospheres’ microbial communities are influenced by a multitude of factors, including but not limited to soil type, environment, and plant genotype. Understanding the impact of these factors on microbial community assembly is key to yielding host-specific and robust benefits for plants, yet it remains challenging. Here, we conducted an artificial ecosystem selection experiment for eight generations of *Arabidopsis thaliana* L*er* and Cvi to select soil microbiomes associated with a higher or lower biomass of the host. This resulted in divergent microbial communities shaped by a complex interplay between random environmental variations, plant genotypes, and biomass selection pressures. In the initial phases of the experiment, the genotype and the biomass selection treatment had modest but significant impacts. Over time, the plant genotype and biomass treatments gained more influence, explaining ~40% of the variation in the microbial community’s composition. Furthermore, a genotype-specific association of plant-growth-promoting rhizobacterial taxa, *Labraceae* with L*er* and *Rhizobiaceae* with Cvi, was observed under selection for high biomass.

## 1. Introduction

The conventional understanding of the host phenotype involves genetics and the environment shaping observable traits. Yet, the last two decades have underscored the microbiome’s significance in shaping the host phenotype, driven by extensive research on its role in ecology, evolution, and host health. The plant microbiome represents a rich source of functional diversity that is not encoded within the host genome. The interactions between the plant host and its microbiome are dynamic and reciprocal, with the plant shaping its immediate environment by exuding specific metabolites, thereby promoting the growth of specific microbial taxa, while the microbiome in turn influences plant health and growth [1]. A multitude of biotic and abiotic factors influence the dynamic nature of the microbiome, including but not limited to plant root exudates [2,3,4]; soil type [5,6,7,8,9]; environment [10]; and various aspects of the plant, including species [11], genotype [4,12,13,14,15] and developmental stage [16,17].

Previous studies in *Arabidopsis thaliana* have shown that the microbial diversity of the soil decreased with proximity to the endophytic compartment, as well as that microbes found in the bulk soil were not enriched in the endophytic compartment, and vice versa [18,19,20]. Furthermore, the microbiomes of the greenhouse-grown plants were similar to those of the field-grown plants, even across different plant compartments—rhizosphere, woody stem, and endophytic compartment—indicating an active role of the plant host in creating and maintaining an environment where certain microbes have improved fitness [21]. The soil environment was a stronger predictor of the rhizosphere’s microbiome structure, though plant genotype has been shown to have a weak influence in some studies [18,20,22]. A follow-up study confirmed that the variation in microbiome communities depends more on the environment than on different *Arabidopsis* varieties and sister species. Though there were some differences between different host species, the core microbiome’s interaction with the plant host at the root endophytic zone was largely consistent and reproducible [23].

Unfortunately, most prior studies were conducted in a single plant growth cycle, precluding an analysis of the temporal stability of the plant–microbiome association. Observations from plant–soil feedback studies highlight that plant–microbiome interactions can be altered by successive growth cycles of a given plant in the same soil [24,25]. Prior studies have employed plant-mediated selection on the soil ecosystem over multiple generations, and this has resulted in consistent effects on plant characteristics such as biomass, flowering time, and germination [26,27]. By selecting for soils where hosts show the desired phenotype, such as high biomass or altered flowering time, it becomes possible to enrich microbes that modulate host traits. Additionally, studies have shown that plants’ responses to abiotic stress, such as drought and salt stress, could be influenced in two ways. The first of these is through generational selection, where plants with desirable stress responses are chosen over multiple generations [28,29,30]. The second occurs by introducing beneficial microbes that are associated with plants and have previously lived in similar drought or salt conditions [31]. Mueller et al. [28] used generational selection to create microbial communities that could improve plant seed production by up to 205%. A soil inoculum obtained from drought-exposed soils improved wheat biomass under drought conditions [31]. Older studies, such as that of Swenson et al. [27], have also shown similar results under optimal growing conditions without necessarily exploring the microbiome. A recent study conducted on the rhizosphere microbiome of wild and domesticated tomato plants over multiple generations demonstrated an escalating influence of the host’s genotype on the microbiome community [32]. A separate study on the phyllosphere microbiome of various tomato genotypes indicated a declining impact of host genotype across successive generations [33].

Despite several earlier efforts, the development of a robust and host-specific microbial community with long-lasting beneficial effects on the plant host remains a challenge. Using an artificial ecosystem selection experiment, we sought to advance our understanding of how the plant genotype, environment, and biomass selection treatment (henceforth referred to as biomass treatment) impact the assembly of host–microbial communities. In this study, we hypothesized that: (i) Genotype, biomass treatment, and environment would play a critical role in microbial community assembly; (ii) generational selection of the soil ecosystem would lead to phenotypes with high and low biomasses; (iii) the influence of plant genotype over multiple generations would result in genotype and biomass-treatment-specific microbiomes. This study enhances our understanding of the temporal dynamics involved in microbial community assembly in the rhizosphere and sets the stage for targeted methodologies establishing resilient plant–microbiome interactions, with potential applications in sustainable agriculture and ecosystem health.

## 2. Materials and Methods

### 2.1. Multi-Generation Selection of Soil Ecosystem

*Arabidopsis thaliana* Cvi and L*er* accessions were cultivated using custom-made “rhizotubes” (Stuart Morey, Univ. of Massachusetts, Boston, unpublished). The tubes were equipped with a black polyethylene sleeve, which effectively blocked light penetration. This design also facilitated the convenient removal of the plants from the pot, granting full access to the root system (Appendix A).

The potting soil used was PRO-MIX PGX, a commercial mixture comprising 80–90% sphagnum peat moss and small quantities of perlite. It was autoclaved twice for 40 min, with a 48 h interval between each sterilization. The sterile potting soil was then sifted through a 3 mm sieve and combined with field soil at a 6:1 ratio for the first generation. The field soil, obtained from the Center for Agricultural Research in Waltham, Massachusetts, was composed of 44% sand, 49% silt, and 7% clay, which is representative of agricultural and grassland ecosystems.

The mixture was homogenized using a custom-made cement mixer and attached to sterile bins to avoid cross-contamination. Distilled water was gradually added to achieve a final ratio of 1:2 (water to soil). The soil was incubated at room temperature for two days following inoculation before the seeds were planted. All seeds used in the experiment were obtained from a single parent plant. Before planting, the seeds were surface-sterilized by treatment with a solution of 50% bleach *v*/*v* and a drop of Tween 20 for 10 min, then rinsed ten times with sterile distilled water. We placed 3–4 seeds in the center of each pot and kept only one seedling per pot after emergence. All plants were grown at 22 °C day/18 °C night with a 12/12 h day–night cycle in a controlled growth chamber. The relative humidity ranged from 35–60%, and the light intensity was 96 µE. Fertilizer was not applied to the plants.

In the first generation of the experiment, 100 plants of each accession (Cvi and L*er*) were grown separately in individual pots. The plants were arranged in a randomized block design to avoid batch effects. All plants were harvested 35 days after germination. The above-ground portion of the plant, the rhizosphere, and the bulk soil from each rhizotube were separated, ensuring that there was no cross-contamination (Appendix A). The above-ground parts of the plant, the stem, and the leaves were dried at 70 °C for 4 days. All of the plants were weighed individually on a closed weighing scale accurate to 1 mg. The root–soil complex (comprising the rhizosphere and the endosphere, henceforth referred to as the rhizosphere) was obtained by shaking the excess soil off the root and placing the root and the remaining attached soil into a sterile 5 mL tube. Bulk soil samples were transferred to a sterile Ziplock bag. Tubes and bags were immediately transferred to dry ice and then stored at −80 °C for DNA analysis.

All subsequent generations were inoculated with a mixture of previously collected bulk soil from the top five and bottom five plants based on above-ground biomass, then combined with sterile potting soil at a ratio of 1:14 (50 gm inoculum:700 gm sterile potting soil). This resulted in two different treatments for each genotype: high-biomass L*er*, low-biomass L*er*, high-biomass Cvi, and low-biomass Cvi. This process was repeated for the remaining seven generations. Starting with the third generation, plants were grown in uninoculated sterile potting soil as a control for random environmental variation (REV). In generation 6, all plants died for unknown reasons. The experiment was then continued using soil from generation 5. Soil sourced from the top six to ten plants and the bottom six to ten plants based on above-ground biomass was used as the inoculum to replant generation 6.

### 2.2. DNA Extraction and 16S rRNA Amplicon Library Prep

Microbial DNA was isolated from the frozen rhizosphere samples using the Machery-Nagel Nucleospin Soil DNA extraction kit (MACHEREY-NAGEL Inc., Allentown, PA, USA). Approximately 0.1 g of the rhizosphere soil sample was used for DNA isolation. All samples were diluted to 5 ng ul^−1^ with PCR-grade water. The 16S rRNA gene was amplified from the isolated DNA samples in triplicate in 96-well PCR plates. The PCR primers used for the 16S rRNA V4 region were 515F (5′-GTGYCAGCMGCCGCGGTAA-3′) [34] and 806R (5′-GGACTACNVGGGTWTCTAAT-3′) [35] for downstream paired-end Illumina (Illumina, Inc., San Diego, CA, USA) barcoded sequencing [36]. The PCR cycling conditions were as follows: 94 °C for 3 min; 25 cycles of 94 °C for 45 s; 50 °C for 60 s; 72 °C for 90 s; and final elongation at 72 °C for 10 min. The triplicate amplified samples were pooled, then purified and normalized using the SequalPrep™ Normalization Plate Kit (Invitrogen Corporation, Carlsbad, ON, Canada). Finally, multiplexed paired-end sequencing was carried out on the Illumina MiSeq (SY-410-1003) platform using earth microbiome project (EMP) primers [37].

### 2.3. Sequence Data Analysis and Statistics

The paired-end sequences obtained from the Illumina MiSeq were demultiplexed with QIIME 2 and converted into individual sequence fastq files for each sample [36,38]. The rest of the sequence processing was carried out in R (Version 4.2.2) using the DADA2 package (Version 1.24.0) [39]. The reads were processed in R using the following command in DADA2: ‘filterAndTrim(fnFs, filtFs, fnRs, filtRs, truncLen = c (145,145), minLen = 50, maxN = 0, maxEE = c (2,2), truncQ = 2, rm.phix = TRUE, compress = TRUE, multithread = TRUE)’. *De novo* OTU (operational taxonomic units) picking was performed using DADA2 (37), which resolves amplicon sequence variants (ASVs) to single-nucleotide differences. Chimeras were removed. Taxonomy was assigned using a 100% cut-off rate for species-level identification with the (2022) GTDB 16S rRNA reference database [40]. The phylogenetic tree was constructed using *IQTree* [41] with 1000 ultrafast bootstraps, and Modelfinder was used to identify the best model. In this study, the best-fit model was SYM + R10, chosen according to BIC. The data comprising the OTU table, phylogenetic tree, taxonomy table, and sample meta-data were then parsed through R studio using the *phyloseq* package [42]. Here, the processing removed OTUs identified as mitochondrial or chloroplast, which had less than five reads across all samples, and samples that had fewer than 8000 reads. The packages *phytools*, *phyloseq*, *microbiome*, and *ggplot2* were used for further data analysis. All sequence data and metadata can be downloaded from PRJNA1010286.

### 2.4. Diversity Analyses

Alpha diversity was computed using two metrics, Observed and the Shannon diversity index. Beta diversity was calculated using weighted and unweighted UniFrac, and subsequently, principle coordinate analysis (PCoA) plots were constructed using the first two axes, which explained the most variance. Linear mixed effect models, via the *lmerTest* package in R, were used to determine changes in alpha and beta diversity over time (lmer (DiveristyMetric, Genotype*Generation*Biomass Treatment, (1|Generation)), with generation as a fixed and random effect. Pairwise comparisons were determined with *emmeans* and adjusted *p*-values were obtained with Tukey’s test.

### 2.5. Adonis

PERMANOVA was carried out for each generation using the adonis2 test. Both weighted and unweighted UniFrac distance matrices were used to account for the abundance, presence/absence, and phylogeny of the microbiome. The following model was used ‘adonis2 (distance.matrix ~ Biomass Treatment + Genotype, data = meta, permutations = 999)’.

### 2.6. Neutral Model

To determine the impact of neutral processes like drift and dispersal, or by deterministic selective forces, such as plant genotype and biomass treatments, on microbial community assembly, we carried out an analysis that was described by Burns et al. [43] to fit microbiome abundance data to the neutral model for prokaryotes from Sloan et al. [44]. The following design was used: sncm.fit (spp = generation (n)_Biomass Treatment, pool = generation (n), stats = T), where n is the nth generation and ‘Biomass Treatment’ represents the high- or low-biomass samples.

### 2.7. Differential Abundance

There are several challenges in estimating differentially abundant (DA) taxa in microbiome data. These include high variability in their abundance, zero-inflated data, and the compositional nature of the data. Nearing et al. [45] compared several methods across 38 datasets and found that different methods often identified different sets of DA taxa. The DA taxa were estimated using ANCOMBC2 [46], a conservative and robust approach. ANCOMBC2 incorporated bias correction, effectively addressing sampling-specific and sequencing biases present in the data. This feature ensures that the analysis is not skewed by any systematic errors introduced during the sampling or sequencing processes. It also conducts a sensitivity analysis for the pseudo-count addition to assess the impacts of different pseudo-count values on zero counts for each taxon. The analysis was run on L*er* and Cvi samples, from OTU to family taxonomic levels, as follows: (fixed effect = (Biomass Treatment + Genotype + Generation), group = Biomass Treatment).

### 2.8. Spearman Correlation for Differentially Abundant OTUs

The ‘cor’ function from the *stats* package (version 4.2.3) was employed to calculate Spearman’s correlation coefficient between differentially abundant OTUs in the high-biomass-treatment samples of L*er*. This was carried out for samples from both the early phase (generation 1 and generation 2) and the late phase (generation 6, generation 7, and generation 8) of the experiment. The results were then visualized using the *corrplot* package (version 0.92). Correlations were computed separately for the early and late phases of the experiment, enabling us to observe changes in the correlation coefficients at different stages of the experiment.

## 3. Results

### 3.1. Changes in Above-Ground Plant Biomass

The ecosystem selection experiment was carried out with respect to the above-ground plant biomass in two different accessions of *Arabidopsis thaliana*—L*er* and Cvi—for eight generations. The two selected treatments were characterized by high and low above-ground biomass. The absolute values for the mean biomass of plants (n = 50) from each genotype and treatment changed substantially through the course of the eight generations of the experiment (Appendix A). Despite the stochastic fluctuations in biomass caused by random environmental variations (REVs), the high-biomass-treatment plants were always the same as or greater than the low-biomass-treatment plants for both genotypes (Figure 1a,b). Due to the significant drop in biomass between generations 1 and 2, an uninoculated control in sterile potting soil was planted in all subsequent generations to serve as a reference for REV. The uninoculated control shows that the drastic drop in biomass after generations 1 and 2 was not a result of the inoculated soil, but was due to other subtle, but crucial, environmental factors that could not be held constant. The above-ground biomass for each generation in terms of deviations from the mean was plotted to give a clearer representation of the difference in phenotype seen in every generation (Figure 2a,b). Significant differences between the low- and high-biomass treatments become apparent from generation 4 onwards. Plants of all treatments in generation 6 died, resulting in a sharp dip, which necessitated repeating that generation (described in Section 2).

### 3.2. Microbial Community Composition

After preprocessing the 16S rRNA metagenomic amplicon reads in DADA 2, a total of 1,897,732 reads with an average of 12,822 reads per sample were obtained. Singletons and chimeras were removed during pre-processing. The most well-represented phyla were *Proteobacteria* (36.2%), *Bacteroidetes* (10.3%), *Planctomycetes* (7.8%), and *Actinobacteria* (7.6%) (Appendix A). The most well-represented classes identified in the dataset were *Alphaproteobacteria* (24.7%), *Gammaproteobacteria* (11.4%), *Bacteroidia* (10.1%), and *Verrucomicrobiae* (6.2%) (Figure 3). This relative abundance is typical of microbiome data, heavily weighted by a few abundant groups and a long tail of rare taxa.

### 3.3. Trends in Alpha Diversity

We used two metrics to assess alpha diversity (Figure 4). Observed (F_7,160_ = 333.054, *p* < 0.001) estimated species richness and showed a significant decrease in the number of OTUs (1477 to ~580) between generation 1 and generation 5, with less pronounced changes in diversity after generation 5. The Shannon diversity index (F_7,160_ = 82.23, *p* < 0.001), which was more sensitive to the difference in abundance, exhibited a comparable trend. Both metrics demonstrated a rapid decline in diversity between generations 1 and 5, with a less pronounced decrease after generation 5. A negative exponential function was fitted to the Observed metric (R^2^ = 0.81, *p* < 0.01) and the Shannon metric (R^2^ = 0.58, *p* < 0.01). The Observed metric displayed a better fit, indicating the increased stability of the microbiome in later generations. In addition, a linear mixed-effect model showed significant interactions between genotype and biomass treatment (F = 8.71, *p* < 0.01); genotype and generation (F =3, *p* < 0.01); and genotype, biomass treatment, and generation (F = 3, *p* < 0.01) for Shannon (Appendix A). No significant interaction terms were found for Observed, which did not consider the abundance of OTUs in its calculations.

### 3.4. Beta Diversity and Principal Coordinate Analysis

Principal coordinate analysis (PCoA) of the weighted UniFrac distances was conducted. They were plotted with the first two axes that captured the most variance in the data, from 73% in generation 1 to more than 90% in generation 8 (Figure 5). The UniFrac distance considers the evolutionary relationships between taxa, which makes it more biologically meaningful than other distance metrics that do not consider the evolutionary history. Weighted UniFrac is more sensitive to differences in the abundance of taxa. The microbiomes associated with the biomass treatments and genotypes were very similar in generation 1, but began to diverge by generation 2. By generation 3, the microbiomes of the L*er* and Cvi genotypes had diverged, and by generation 5, the high- and low-biomass treatments appeared to have further distinguished the microbial communities. The divergence of the microbial community in generations 4 and 5 caused by the high/low biomass treatments also aligned with the onset of notable differences in the above-ground plant biomass within these generations.

The complex interplay between genotype and biomass treatment was modeled using a PERMANOVA test with both weighted and unweighted UniFrac distances to account for differences in abundance and presence or absence of OTUs, respectively. The resulting R^2^ values for genotype and biomass treatment and residuals as a proxy for REVs were plotted (Figure 6a,b). For both the weighted and the unweighted UniFrac, the variance in the microbial community explained by both genotype and biomass treatment increased significantly over the eight generations. Despite observing a decrease in the variance accounted for by the residuals, which could potentially signify a decline in the influence of REVs, they accounted for ~50 percent of the variability within the microbial community in generation 8. For all metrics, weighted UniFrac exhibited more fluctuations compared to the unweighted UniFrac. The weighted UniFrac analyses found that genotype (F_1,39_ = 1.55, *p* < 0.05) and biomass treatment (F_1,39_ = 1.33, *p* < 0.001) explained 5.7% and 13.7%, respectively, of the dissimilarity between microbiomes in generation 1. By generation 8, the genotype (F_1,14_ = 5.14, *p* < 0.05) and biomass treatment (F_1,14_ = 5.48, *p* < 0.001) explained 24% and 22%, respectively, of the dissimilarity between microbiomes (Appendix A). For unweighted UniFrac, the influence of genotype on the microbial community increased from 3.3% in generation 1 (F_1,39_ = 1.33, *p* < 0.01) to 26% in generation 8 (F_1,39_ = 5.13, *p* < 0.001), and for biomass, it increased from 3.8% in generation 1 (F_1,14_ = 1.55, *p* < 0.001) to 13% in generation 8 (F_1,14_ = 2.58, *p* < 0.05) (Appendix A).

### 3.5. Sloan Neutral Model

Many forces can alter a microbial community’s structure and dynamics. For our study, these can be divided into selective forces, such as the plant genotype being colonized and the biomass selection process, or neutral, stochastic forces like dispersal that are innate in any experiment or environment. The Sloan neutral model was fit to the data to assess the importance of selective versus neutral drivers of change in the experiment. If neutral processes were the driving force in the microbial community assembly, the null hypothesis would be that all generations of the experiment would fit the model equally well. The model fit to neutrality represented by the R^2^ value decreased from 0.795 in generation 1 to 0.59 in generation 8, demonstrating the increasing importance of selective drivers over the course of the experiment (Figure 7).

### 3.6. Differential Abundance and Correlation Analyses

Differentially abundant (DA) taxa that distinguished the high-biomass treatments from the low-biomass treatments were determined using ANCOMBC2 at the family level. Taxa lacking classification at the family level were denoted by the preceding identifiable taxonomic tier. The OTUs that were present in higher (red) and lower (blue) abundances in the high biomass treatment were identified (Figure 8a,b). Among the DA taxa in this study, several are known to benefit plants. Bacteria within the class *Bacilli* promote plant growth [47]. Members of the class *Gemmatimonadetes* have been shown to exhibit a positive association with vegetation restoration, plant richness, and soil nutrients [48]. *Cytophagacea* are chemoorganotrophs, important for remineralizing organic materials into micronutrients. They could support both mycelial growth and plant nutrition [49].

*Enterobacteriaceae*, *Paenibacillaceae*, and *JACDCH01* were all present in higher abundances in the high-biomass treatments of both L*er* and Cvi. Most *Paenibacillaceae* members predominantly inhabited soil, frequently in close association with plant roots [50]. These rhizobacteria are known to play a significant role in enhancing plant growth, and they possess potential applications in agriculture. *Enterobacteriaceae* has been reported to enhance plant growth [51,52,53].

Additionally, *Order_NS11-12g*, *BJHT01*, *JACDCH01*, and *UBA6156* were more abundant in Cvi under high-biomass conditions, but less abundant in L*er* under high-biomass conditions. The majority of these were uncultivated or candidate taxa. Two alternative families in the order *Rhizobiales* appeared to be differentially associated with the two plant genotypes: *Labraceae* with L*er* and *Rhizobiaceae* with Cvi [54]. These could indicate genotype-specific interactions with different members of the microbial community.

The relative abundances of all OTUs that exhibited higher abundance (enriched) in samples from the high-biomass treatment in L*er* and Cvi are depicted in Figure 8c and Figure 8d, respectively. These OTUs consistently exhibited an increasing trend of relative abundance across successive generations. A linear model was employed to test this trend, resulting in R-squared values of 0.57 (*p* < 0.01) for L*er* and 0.73 (*p* < 0.01) for Cvi. Furthermore, OTUs that were lower in abundance (depleted) in samples from the high-biomass treatment of L*er* and Cvi are depicted in Figure 8e and Figure 8f, respectively. These OTUs persisted in low abundance across all generations, suggesting a lack of selective pressure on this group of microbes in the samples which received the high-biomass treatment.

We then conducted a Spearman’s correlation analysis using 58 OTUs identified as enriched in high-biomass-treatment samples of L*er* (Figure 9a,b). This analysis was carried out for samples from both the early phases (generations 1 and 2) and the late phases (generations 6, 7, and 8) of the experiment. Our observations indicated that, during the early phases, the associations between OTUs were comparatively weaker than during the later phases, suggesting a trend of OTUs being selected in groups rather than individually. Additionally, we presented specific examples of OTUs that showed significant differences in the number of pairwise positive and negative associations between the early and late phases of the experiment (Figure 9c). This visualization highlights trends spanning both the early and late phases of the experiment, with certain OTUs demonstrating an increase in positive pairwise associations and a decrease in negative pairwise associations as generations progressed. Notable examples include *OTU1606_Chitinophagaceae*, *OTU162_Bdellovibrionaceae*, *OTU94_Class_Bacilli*, and *OTU359_Order_AKYH767*. Conversely, others displayed the opposite trend, such as *OTU763_Fimbriimonadaceae*, *OTU142_Sphingobacteriaceae*, and *OTU1005_Labraceae*.

## 4. Discussion

We applied artificial ecosystem selection to the eight generations in *Arabidopsis thaliana* L*er* and Cvi to select soil microbiomes associated with the higher or lower biomass of the host. In contrast to some previous studies [26,32], we did not fertilize plants, thus maintaining nutrient limitation and thereby promoting the interaction of the plant with the microbiome [55]. Over the course of eight generations, a response to the selection, most apparent after generations 4 and 5, was evident in both the gradually shifting microbiome composition and the plant biomass. The microbiome selection process noticeably influenced the plant biomass despite large phenotypic variation from one generation to the next. Stochasticity in growth through generations may be attributed to random environmental variations (REVs), as indicated by Swenson et al. [27]. This was observed in the uninoculated sterile potting soil reference, which showed similar patterns of variability to the inoculated treatments across generations (Figure 1). Ecological variability among generations, even under the conditions of a controlled growth chamber, is a common characteristic of similar studies [56], and may be due to minute fluctuations in the growth chamber conditions or the batch and age of the potting soil.

We observed a gradual decline in microbial species richness during the initial generations of the experiment. This was evident from the observed OTU counts, as well as from the Chao1 and Shannon diversity indices. This pattern is consistent with findings from other studies [31,32,33], highlighting the impact of selection pressures as the microbial communities adapted to the host plant’s environment. However, in our study, which continued for about twice the number of generations as these earlier studies, we observed a stabilization of richness and alpha diversity in the later generations. This was further substantiated by the stronger fit to the negative exponential function when compared to a linear model, indicating constant and proportional changes in alpha diversity and a trend of the rhizosphere microbiome towards reduced diversity loss, accompanied by heightened stability, over eight generations. Previous studies have suggested that the stability in alpha diversity metrics in later generations may possibly be due to the retention of microbiome community members through robust selection and improved fitness with the host [57,58,59]. Over time, less fit individuals are outcompeted by microbes that have better fitness in the host environment [59,60].

To further understand the complex interplay between the forces of directed selection (plant genotype and biomass treatment), we conducted a PCoA. Over successive generations, the results demonstrated a strengthening effect of the plant genotypes and biomass treatment on the microbial community (Figure 5). This result was modeled using PERMANOVA for both weighted and unweighted UniFrac distances. Both the weighted and unweighted UniFrac showed marked increases in the proportion of variance explained by genotype and biomass treatment, with the unweighted UniFrac exhibiting larger increases from generation to generation (Figure 6). Both metrics considered phylogenetic relatedness. Weighted UniFrac considers the absolute abundance of OTUs, and is generally used to study changes in microbial community structure. Unweighted UniFrac considers only the presence/absence of OTUs, and is generally used to study changes in microbial community composition. The results suggest that the variance in the microbiome’s composition (presence/absence of OTUs), as explained by genotype, consistently increases, whereas the variance in the microbiome structure (considers the abundance of OTUs), as explained by genotype, greatly fluctuates with changes in abundant taxa.

Here, we would like to highlight that pronounced shifts in biomass between the high/low biomass treatments, alpha diversity, and beta diversity all occur around generations 4 and 5 (Figure 1 and Figure 2). This marks the point at which the changes in alpha diversity stabilize (Figure 4) and the divergence between microbial communities by high/low biomass treatments in the PCoA plots becomes more pronounced (Figure 5). It is possible that, over the first four to five generations, the initial microbial community underwent a period of restructuring before it stabilized and formed four distinct communities under selection by high/low biomass treatments and by plant genotype. We propose that a complex interaction between plants and their associated microbiome ensued, where differences in root exudation patterns between the two genotypes presumably established associations with microbes in the early generations. Simultaneously, during this period, the biomass treatment likely promoted host-specific, microbe-mediated interactions that modulated plant biomass. This restructuring of the microbial community was driven by both structural (abundance of OTUs) and compositional (presence/absence of OTUs) effects in the microbiome (Figure 6a,b). Differences in the abundance of taxa driven by the biomass treatment and genotype selection pressures were a larger contributor to the generation-to-generation variation during the experiment.

Changes in the assembly of microbial communities can arise from either selective pressure, like the plant’s genotype or the biomass treatment, as observed in this experiment, or stochastic processes, like minute changes in the growth chamber humidity or microenvironment. To gain deeper insights into the influence of selection on microbial community assembly, a neutral model was fit to the data [43,44]. Interestingly, this revealed a progressive decline in fit to neutrality, indicated by decreasing R^2^ values and increasing AIC values (Figure 7). This suggests an increasing influence of selective forces such as biomass treatment and genotype over multiple generations of ecosystem selection. This has also previously been observed by Morella et al. [33].

Previous studies in *Arabidopsis thaliana* have often detected only a weak genotypic effect on the rhizosphere microbiome [18,23,61]. However, a majority of these studies are focused on a single growth cycle. Insights from plant–soil feedback research emphasize that the interactions between plants and their microbiomes can be modified by consecutive growth cycles of the same plant species in its respective soil [24,25]. Recent research has presented conflicting findings on the influence of the host genotype [32,33,62]. The different experimental designs, communities being assessed, and the length of these experiments, alongside the high stochastic environmental variation that is clearly associated with these microbial studies, likely account for the conflicting findings. In our investigation, we found that during the initial stages, genotype and biomass treatments had modest, but significant, impacts. Over time, the plant genotype and biomass treatments exerted an increasing influence. Together, these selective forces explained ~40% of the variation in the microbial community’s composition in these later generations.

A key aim of microbiome engineering has been to develop host-specific microbial communities that impart lasting beneficial effects to the plant host [26,28,31,63]. Herein, we show the enrichment of some common taxa in high-biomass treatments, but also some genotype-specific changes. Despite starting from the same soil, only three common families were enriched in the high-biomass treatment samples of both the L*er* and Cvi genotypes. In addition, well-known plant-growth-promoting rhizobacteria were enriched in a genotype-specific manner for both the L*er* and Cvi samples that underwent the high-biomass treatment. In the order *Rhizobiales*, *Labraceae* was enriched in L*er*, while *Rhizobiaceae* was enriched in Cvi [54]. Moreover, in high-biomass conditions, Cvi had increased levels of the taxa *Order_NS11-12g*, *BJHT01*, *JACDCH01*, and *UBA6156*, while L*er* had decreased levels of these. These data suggest genotype-specific plant genetic control of the rhizosphere microbiome.

To further explore how the abundance of these OTUs changed across successive generations, we grouped all the DA OTUs that were enriched or depleted in the samples with the high-biomass treatment and calculated the sum of their relative abundances. The OTUs enriched in the samples of high biomass treatment displayed an increasing trend of relative abundance over subsequent generations (Figure 8c,d). A linear model was fitted to these data with high R^2^ values, reinforcing the notion that the genotype and biomass treatments exerted selective pressure on entire microbial groups rather than individual species. Conversely, the OTUs that were depleted in the samples which underwent high-biomass treatment consistently remained in a state of low relative abundance throughout the entirety of the experiment (Figure 8e,f). This observation may indicate that the selective pressures exerted by the genotype and biomass treatments do not influence this particular group of microbes. 

The results of the correlation analysis revealed a strong directional shift in the relationships between the OTUs in the later phase of the experiment compared to the early phase of the experiment. This analysis provides further empirical evidence that the OTUs were selected in groups. This is particularly evident for members of *Chitinophagaeae*, which showed a strong increase in pairwise positive associations and a concomitant decrease in negative associations. Strengthening of microbial associations has been observed in previous succession studies on different plant species, including *Arabidopsis thaliana* [8,64,65]. Luo et al. also found stronger positive associations in the rhizosphere, highlighting the importance of this zone in promoting stronger microbial associations [64].

Taken collectively, these findings suggest that the presence of diverse selection pressures influences the rhizosphere microbiome. Initially, it seems that neutral processes seem to play a major role in determining the structure and composition of the microbiome (Figure 7). As generations progress, we observed an increase in the strength of directional selection, possibly driven by genotype and biomass treatment, resulting in the enrichment of specific groups of microbes, including some PGPRs (Figure 8c,d). The persistence of depleted microbes in the high-biomass-treatment samples suggests the possibility of negative frequency-dependent selection preserving them at a low abundance. On the other hand, it is possible that this group did not experience any observable selection pressure. 

Multiple studies have shown variations in the structure of the rhizosphere microbiome, even within closely related plant genotypes, highlighting the importance of genotype-specific root exudates in forming associations with the corresponding microbiome [66,67,68,69,70,71,72]. While previous studies have observed qualitative distinctions in the root exudate profiles of L*er* and Cvi, these profiles lack quantitative characterization [67]. Currently, our understanding is limited to some characterization of root exudates for the L*er* [67,68,73], and to the best of our knowledge, there has been no in-depth characterization of Cvi’s root exudate profile. Closer associations in the rhizosphere are formed between plants and bacterial partners capable of metabolizing specific compounds. In a previous *Arabidopsis thaliana* study, it was observed that rhizosphere-enriched microbes, such as *Terracoccus* sp. 273, excelled in utilizing SA as a carbon source. Conversely, microbes like *Mitsuaria* sp. were unable to do so and were found in lower abundance [61]. Other plants, such as maize, release benzoxazinoids (BXs) in their root exudates, and there is a negative correlation in terms of abundance between *Glomeromycota* members and BXs [24,74,75]. It is important to emphasize that microbes are also able to reciprocally influence the root exudation patterns of plants. Pea plants treated with *Pseudomonas aeruginosa* PJHU15 and *Bacillus subtilis* BHHU100 showed a significant increase in gallic acid and other phenolics compared to control plants [76]. Furthermore, in response to *Pseudomonas syringae* pv *tomato* infection, *Arabidopsis thaliana* roots shift their exudation patterns, releasing more malic acid. Malic acid is a chemoattractant, selectively inducing *Bacillus subtilis* to bind to the infected plant’s roots and, thus, improving plant disease resistance [77]. Future explorations should prioritize profiling root exudates, along with comparative metagenomics, metatranscriptomics, and metabolomics studies. This will play a role in decoding chemical crosstalk, which could eventually foster reproducible interactions between plants and microbes, leading to an overall enhancement in plant fitness.

The relationship between plants and microorganisms in the rhizosphere involves complex and diverse interactions which influence crucial ecological and physiological processes. These interactions can be mutualistic, competitive, or antagonistic. Many biotic and abiotic factors act in concert to influence the dynamic rhizosphere microbiome. Gaining insight into how these factors influence the assembly of microbial communities is crucial for obtaining targeted and long-lasting advantages for plants. Nevertheless, establishing a strong and host-specific microbial community that consistently provides beneficial effects continues to be a persistent challenge. We have shown that, despite stochastic fluctuation due to REVs, it is possible to select for microbial communities that impact biomass in a genotype-specific manner within four generations, addressing this key challenge in microbial community engineering [31,63]. The rhizosphere microbiome that evolves under plant-mediated selection has the potential to offer improved survivability and efficacy when applied as an inoculum to the plant [78,79]. Multi-species ecosystem selection provides a perspective on microbiome assembly that indicates the holobiont, and the evolving relationship between the host and the microbiome contributes to emerging properties beyond those predicted by the host’s genotype or the initial composition of the microbial community. This study enhances our understanding of the temporal dynamics involved in microbial assembly in the rhizosphere, and has implications for sustainable agriculture, evolution, and ecology.

## 5. Conclusions

The detrimental impact of agrochemicals has emphasized the importance of adopting sustainable agricultural practices [80,81]. Microbial formulations of PGPRs as inoculants may boost soil fertility, promote favorable plant phenotypes, and reduce the harmful effects of agrochemicals. However, numerous challenges remain in terms of transitioning from the lab to the field—in particular, the survivability and efficacy of these individual strains [82,83,84]. Ecosystem selection overcomes these challenges by selecting for entire groups of microbes rather than individuals, mitigating the potential loss of function due to the extinction of essential microbes. We utilized multi-generational ecosystem selection with respect to above-ground plant biomass for eight generations in two common inbred genotypes of *Arabidopsis thaliana*: L*er* and Cvi. The composition of the resulting rhizosphere microbiome was shaped by a complex interplay between environmental factors, plant genotype, and biomass treatment. In the first few generations of the experiment, the genotype and biomass treatments played a modest role in shaping the microbial community. Over time, the plant genotype and biomass treatments exerted increasing influence, explaining ~40% of the difference in the microbial community composition. We successfully cultivated microbiomes that were associated with either increased or reduced plant biomass, and an enrichment of known plant-growth-promoting rhizobacteria (PGPR) was observed with the high-biomass treatment. The microbiomes identified in this study alter plant biomass and growth, presumably by altering the availability of nutrients in the soil or by mitigating biotic/abiotic stressors.

Translating the findings of such studies from a lab to a field setting requires further optimization of the inoculum transfer [28,85]. For instance, Wright et al. made use of artificial selection to select for active degraders of chitin by timing the harvest window to collect inoculum at the same time that the desired trait (chitinase activity) was at its peak and not at a pre-defined incubation time [86]. However, the dynamic nature of rhizosphere microbiomes, where member abundances constantly shift based on the host plant’s genotype and developmental stage, as well as environmental factors, adds complexity to the challenge of identifying the optimal time point for transfer in plant microbiome studies. Future research should focus on the meticulous refinement of transfer timing, storage methods, and application techniques for the selected microbial inoculum to achieve sustainable agricultural practices and improved soil health through ecosystem selection.

## Figures and Tables

**Figure 1 microorganisms-11-02932-f001:**
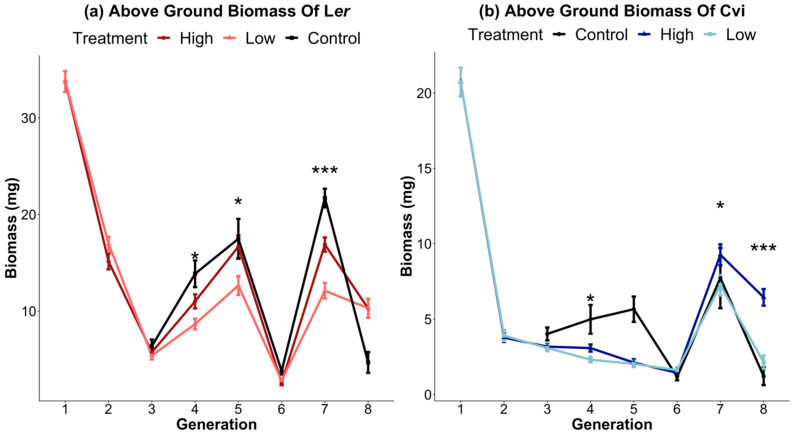
Ecosystem selection leads to changes in above-ground plant biomass. Above-ground biomass for genotypes L*er* (**a**) and Cvi (**b**) of *Arabidopsis thaliana*. Dark lines represent the high-biomass treatment, and light lines represent the low-biomass treatment. Each point represents the mean of n = 50 microcosms. The mean biomass of the selection lines differed for several generations of both genotypes (*t*-test *p*-value * < 0.05; *** < 0.001).

**Figure 2 microorganisms-11-02932-f002:**
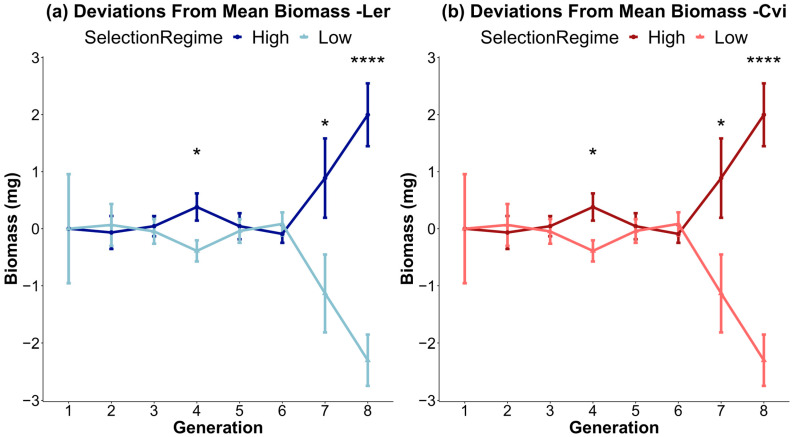
Above-ground biomass represented as deviation from the mean for each generation. By plotting deviations from the mean, the influence of REVs on above-ground biomass is mitigated for genotypes L*er* (**a**) and Cvi (**b**). The dark lines denote high-biomass treatments, while the light lines represent low-biomass treatments. Each data point represents the average of n = 50 microcosms (*t*-test *p*-value * < 0.05; **** < 0.001).

**Figure 3 microorganisms-11-02932-f003:**
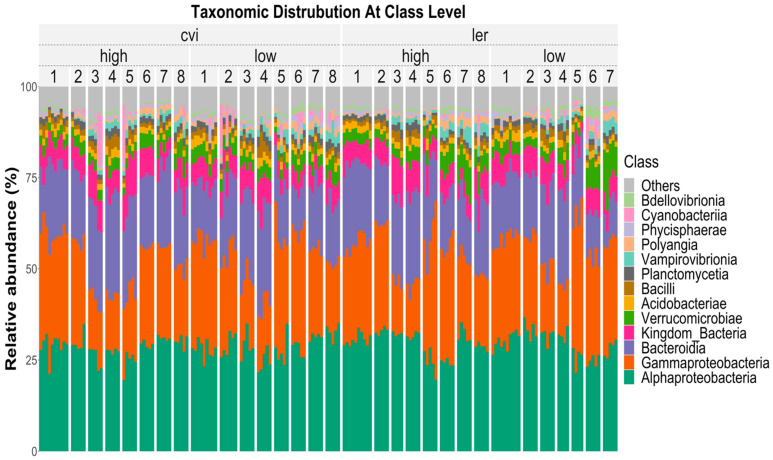
Taxonomic distribution of the microbial community is represented in terms of the relative abundance at the Class level. The plot shows eight generations (1–8) of the high- and low-biomass treatments of both the Cvi (left) and Ler (right) genotypes of Arabidopsis thaliana. Ten samples were assayed for the generation 1 experiments, but only five for all subsequent generations. The most represented classes across all samples were Alphaproteobacteria (24.7%), Gammaproteobacteria (11.4%), Bacteroidia (10.1%), and Verrucomicrobiae (6.2%).

**Figure 4 microorganisms-11-02932-f004:**
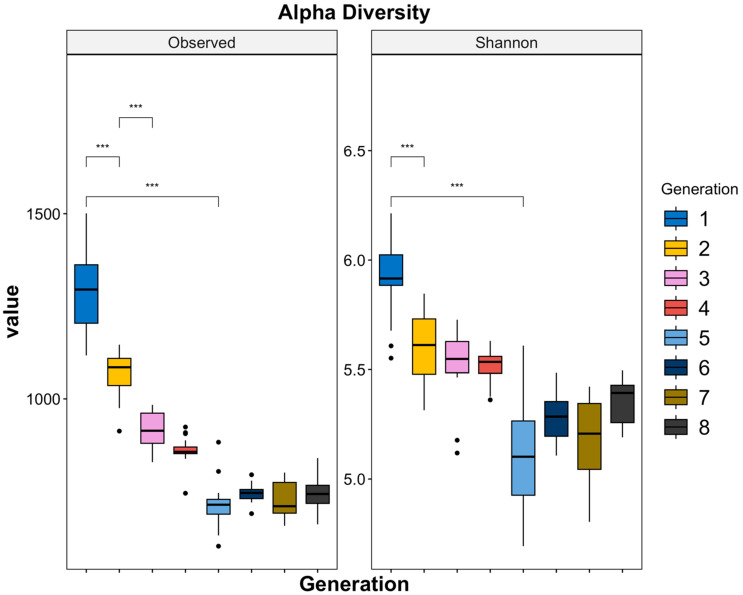
Changes in alpha diversity. Two alpha diversity indices were computed, including the Observed and Shannon metrics. Both metrics demonstrated a rapid decline in diversity between generations 1 and 5, with a less pronounced decrease after generation 5. Additionally, a negative exponential function was fit to the Observed (R^2^ = 0.81, *p* < 0.01) and Shannon (R^2^ = 0.58, *p* < 0.01) metrics, with Observed showing a better fit, indicating the increased stability of the microbiome in later generations (Wilcox test, *** < 0.001).

**Figure 5 microorganisms-11-02932-f005:**
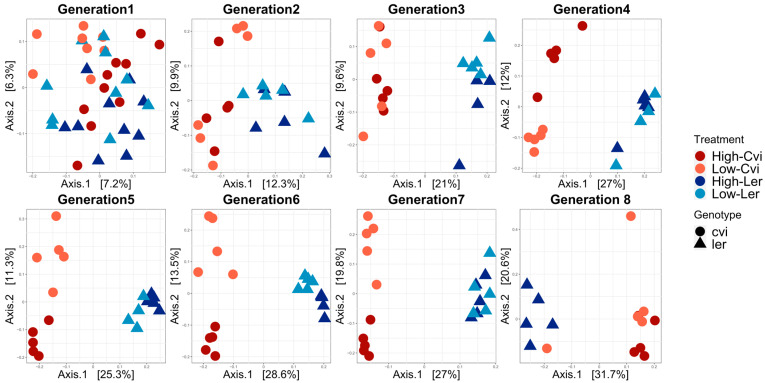
Host genotype and biomass treatment shape the composition of the microbiome. Principle coordinate analysis (PCoA) was conducted using unweighted UniFrac for generations 1 through 8. Each point represents an individual sample, with microbial communities defined as Cvi, represented by red/orange circles, and L*er*, represented by dark/light blue triangles. The first two coordinate axes to be plotted accounted for the highest variation in the data, ranging from 13% in generation 1 to approximately 52% in generation 8. While there was no discernible clustering in generation 1, both genotypes and high/low biomass treatments exhibited greater clustering over the course of the experiment.

**Figure 6 microorganisms-11-02932-f006:**
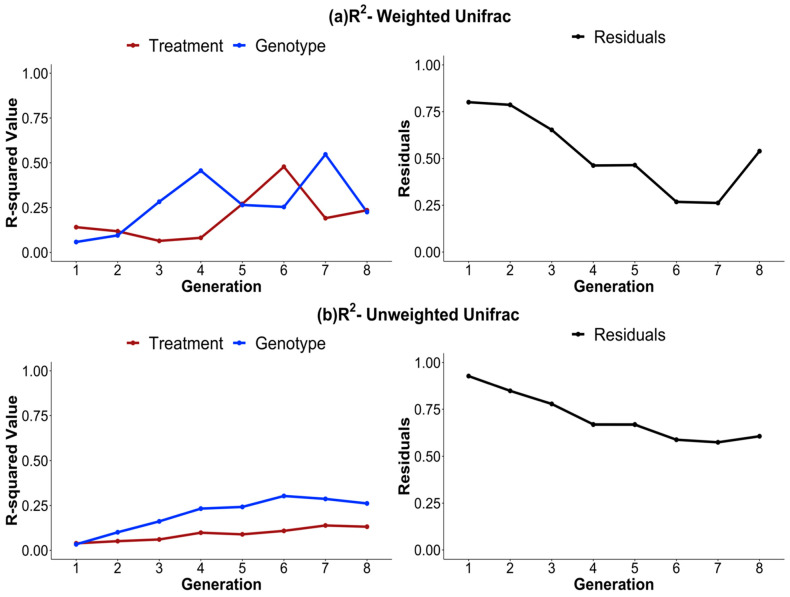
Variance in microbiome structure and composition, explained by genotype and biomass treatment increases over generations. The R^2^ values were computed using the PERMANOVA test with the adonis2 function for generations 1 through 8 for (**a**) weighted and (**b**) unweighted UniFrac distances. The resulting R^2^ and variance explained by residual values for the biomass treatment (brown), genotype (blue), and residuals (black) are plotted. Residuals are plotted as a proxy for random environmental variations (REVs). There was a marked increase in the influence of both genotype and biomass treatment on differences in the microbial community from generations 1 to 8. Despite observing a decrease in the variance accounted for by the residuals, which could potentially signify a decline in the influence of REVs, the model elucidated more than 50 percent of the variability within the microbial community in the 8th generation. These differences are more pronounced in the weighted vs. the unweighted UniFrac model (distance.matrix ~ biomass treatment + genotype).

**Figure 7 microorganisms-11-02932-f007:**
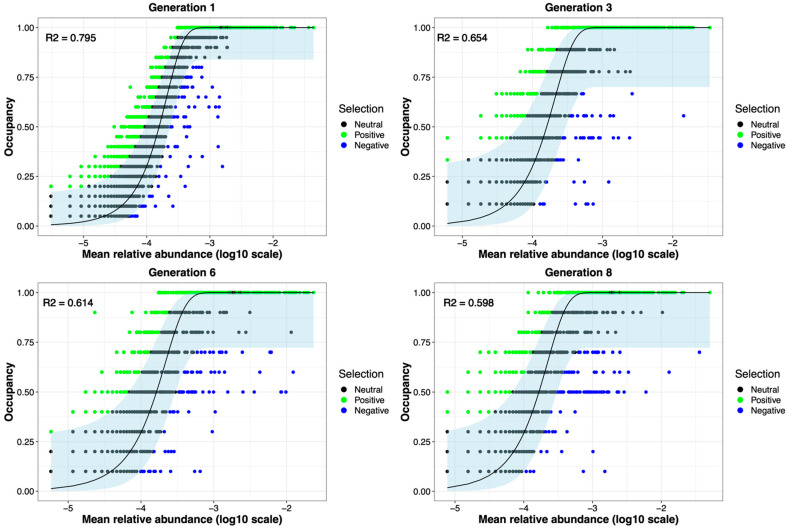
Fit of the neutral model diminished as generations progressed. The occupancy (prevalence of OTU in samples), plotted against log_10_ abundance, is depicted using the Burns model. OTUs that are neutral, i.e., not selected for or against, are indicated in black. Green- and blue-colored OTUs signify positive and negative selection, respectively. This plot highlights the presence of numerous microbes that underwent directed selection throughout the experiment.

**Figure 8 microorganisms-11-02932-f008:**
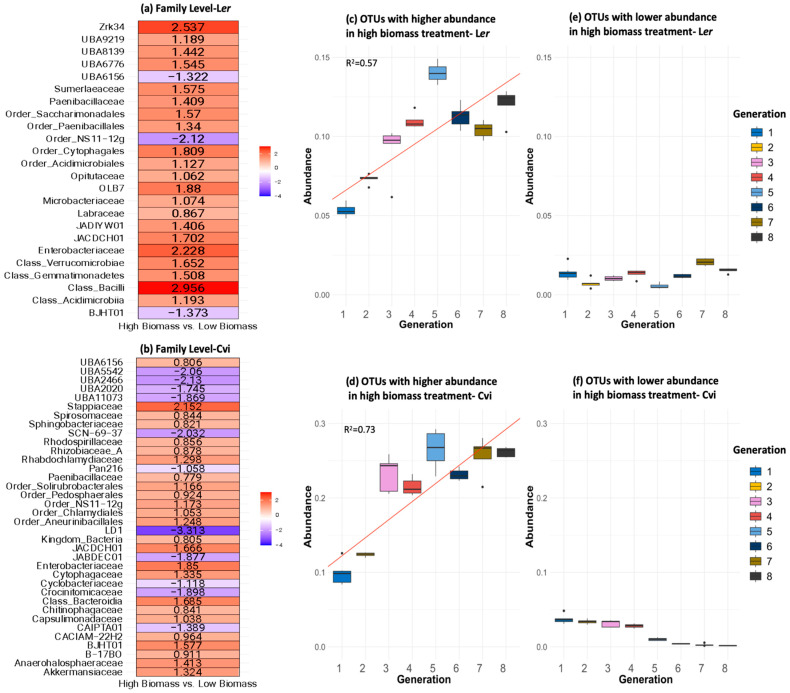
Differentially abundant OTUs. ANCOMBC2 results are illustrated as log_2_ fold changes at the family level for high-biomass vs. low-biomass samples. Each taxon listed in the figure represents a single OTU in either the L*er* (**a**) or Cvi (**b**) microbial community, identified to the family level if possible. The scale indicates log_2_ fold changes, with red and blue representing positive and negative fold changes, respectively, in the high-biomass samples. Well-known plant-growth-promoting bacteria, such as *Paenibacillaceae*, *Bacilli*, *Labraceae*, *Rhizobiaceae*, and*Bdellovibrio* were present in higher abundance. A vast majority of the other taxa identified were uncultivated, and there is little to no information on them. (**c**,**d**): Box plots of relative abundance for all OTUs that were enriched in high-biomass treatment samples of genotypes L*er* and Cvi, respectively, with colors indicating different generations. These OTUs demonstrated a consistent trend of increasing relative abundance across successive generations. To test this trend (red line), a linear model was fitted to the data, resulting in an R^2^ value of 0.57 (*p* < 0.01) for L*er* and 0.73 (*p* < 0.01) for Cvi. (**e**,**f**) are box plots of relative abundance for all OTUs that were depleted in samples from the high-biomass treatment in genotypes L*er* and Cvi, respectively, and are colored according to generation.

**Figure 9 microorganisms-11-02932-f009:**
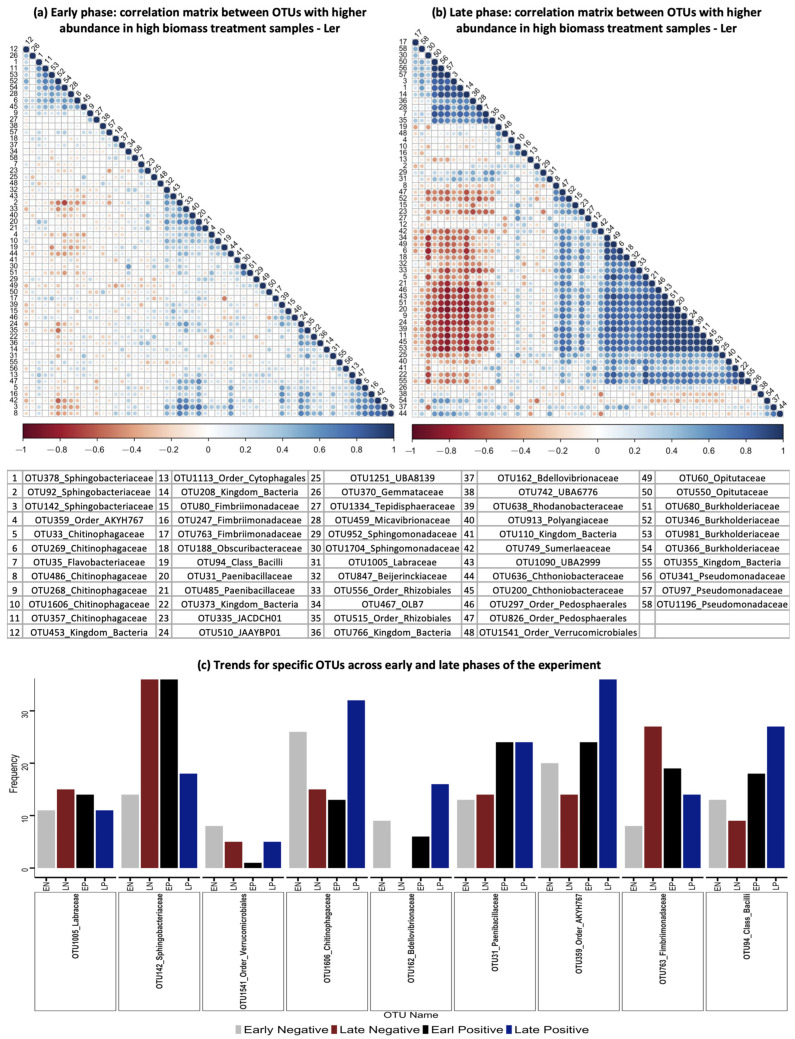
Pairwise spearman correlation analysis for differentially abundant OTUs. Spearman correlation results are illustrated between OTUs enriched in the high-biomass-treatment samples of the L*er* genotype in the early phase (generation 1 and generation 2) (**a**) and late phase (generation 6, generation 7, and generation 8) (**b**) of the experiment. The color gradient depicts the correlation coefficient’s strength for each OTU pair, from −1 (dark red) to 1 (dark blue). The legend represents a key, where each number corresponds to the OTU name and taxonomy at the family level, when available. (**c**): Barplot of a subset of OTUs. The bars represent OTUs, along with the number of pairwise positive or negative associations. This visualization highlights trends over the early and late phases of the experiment, with some OTUs exhibiting increases in positive pairwise associations (*OTU1606_Chitinophagaceae*, *OTU162_Bdellovibrionaceae*, *OTU94_Class_Bacilli*, and *OTU359_Order_AKYH767*) as generations progressed and decreases in negative pairwise associations, while others exhibited the opposite trend (*OTU763_Fimbriimonadaceae*, *OTU142_Sphingobacteriaceae*, and *OTU1005_Labraceae*).

## Data Availability

All sequence data and metadata can be downloaded from PRJNA1010286.

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
