# Peer review of "Multi-Generation Ecosystem Selection of Rhizosphere Microbial Communities Associated with Plant Genotype and Biomass in Arabidopsis thaliana"

_microorganisms, 2023, doi:10.3390/microorganisms11122932_

Round 1

Reviewer 1 Report

Comments and Suggestions for Authors

The paper provides valuable insights into the dynamic interplay of plant genotype, environment, and biomass selection on rhizosphere microbial communities, laying the groundwork for future research and practical applications in agriculture and ecology. The presentation of plant biomass dynamics results, and microbial community composition are clear and well-supported with figures, providing a comprehensive understanding of the changes in above-ground plant biomass over eight generations. The comprehensive analysis of alpha and beta diversity, utilizing multiple metrics and statistical approaches, and integration of selective and neutral models helped elucidate the forces driving microbial community changes.

However, there are several areas that must be improved:
1- Provide explicit values or ranges for above-ground biomass in each generation for clarity.

2- Thoroughly explain the biomass dip in generation 6.

3- Consider additional analyses, such as correlation or network analysis, to explore microbial taxa-plant biomass relationships.

4- Offer a more detailed discussion on the stabilization of microbial diversity in later generations.

5- Integrate key findings from the results section into the conclusion for a cohesive narrative.

6- Please discuss root exudate profiles/roles in conjunction with microbial community dynamics, and their (possible) role through the generations in shaping differentially abundant microbial taxa in high biomass conditions.

7- Mention briefly (perhaps at the end of the discussion session) practical implications for agriculture or ecosystem management based on the identified microbial communities. How could the identified microbial communities be harnessed for sustainable agriculture or environmental restoration?.

Comments on the Quality of English Language

Minor editing (typos) of English language required

Author Response

Dear Reviewer 1,

Thank you for your valuable time in reviewing this manuscript. Your comments are greatly appreciated. We have addressed your comments below.

1- Provide explicit values or ranges for above-ground biomass in each generation for clarity.

Thank you for your valuable time in reviewing this manuscript. As per your recommendation, we have added a table in the supplementary showing a range of biomass (mg) – mean and standard deviation for each generation (Supplementary Table S10 and S11).

2- Thoroughly explain the biomass dip in generation 6.

Thank you for your recommendation. Please refer below for an explanation of the biomass dip in this generation 6.

After seeing the low biomass in all plants, including the sterile potting control samples in generation 6, we contacted the authors, who have previously conducted similar experiments in Arabidopsis thaliana, Jenny-Kao Kniffin and David Sloan Wilson, both of whom suggested continuing the experiment as described in the methods (lines 131-134). By using inoculum obtained from the top and bottom sixth to the tenth plants by biomass from generation 5, generation 6 plants were regrown in a fresh batch of autoclaved potting soil. Upon regrowing the plants, we still observed very small plants; however, the second time the plants were grown, they did not die. The biomass of all the plants was measured using a fine scale, and the soil from the top five and bottom five plants by biomass was used as an inoculation for generation 7, where we observed a reversion back to the previously observed range in biomass.

In our study, we included a sterile control (autoclaved for 3 rounds with a 48-hour gap at room temperature), indicating that the cause of all plants dying in generation 6 was not the inoculation with microbes but some other factor, which unfortunately could not be determined.

This phenomenon of sudden, widespread plant death was also observed in generation 15 of a previous experiment conducted by Swenson et al [reference 28]. Here, they restarted the experiment in a similar fashion as mentioned above and saw a recovery to their previously observed range in biomass in generation 16.

3- Consider additional analyses, such as correlation or network analysis, to explore microbial taxa-plant biomass relationships.

Thank you for your valuable insight. Your recommendation has greatly added value to the paper. In accordance with your suggestion, we conducted Spearman correlation analyses on differentially abundant OTUs enriched in the high biomass treatment samples for Ler at the family-level resolution for taxonomy. This analysis was performed for samples in both the early phase (generations 1 and 2) and the late phase (generations 6, 7, and 8).

4- Offer a more detailed discussion on the stabilization of microbial diversity in later generations.

Thank you for your valuable insight. We have updated the discussion section on the stabilization of microbial diversity, with additional references.

5- Integrate key findings from the results section into the conclusion for a cohesive narrative.

Thank you for your recommendation. In accordance with your suggestion, we have incorporated a conclusion section that synthesizes the findings of our research for a cohesive narrative. 

6- Please discuss root exudate profiles/roles in conjunction with microbial community dynamics, and their (possible) role through the generations in shaping differentially abundant microbial taxa in high biomass conditions.

Thank you for your input. We have updated the discussion section to include the potential role of root exudates in shaping differentially abundant microbes. Your guidance is highly appreciated. 

7- Mention briefly (perhaps at the end of the discussion session) practical implications for agriculture or ecosystem management based on the identified microbial communities. How could the identified microbial communities be harnessed for sustainable agriculture or environmental restoration

Thank you for your recommendation. In accordance with your suggestion, we have incorporated a conclusion section that discusses the practical applications for sustainable agriculture.

Reviewer 2 Report

Comments and Suggestions for Authors

The paper is interesting and makes a potential contribution in the field of knowledge of microbial activity in the soil. Most of the chapters are well done, they require finishing, but also some minor improvements in the part of international understanding of the presented research:

- in my opinion, the research hypothesis and research objectives must end the introduction chapter; here the authors put the part of conclusions (lines 74-86) which can also be found in the abstract.

- in the material and method, the soil used is presented only in terms of texture (line 100-101), additional information on pH, organic matter content, etc., may be useful.

- Bulk soil samples (line 122) ...what size did you use?

- don't you think it would be useful to synthesize some conclusions?

Comments on the Quality of English Language

Minor editing of English language required.

Author Response

Dear Reviewer 2,

Thank you for your valuable time in reviewing this manuscript. Your comments are greatly appreciated. We have addressed your comments below.

- in my opinion, the research hypothesis and research objectives must end the introduction chapter; here the authors put the part of conclusions (lines 74-86) which can also be found in the abstract.

Thank you for your valuable time in reviewing this manuscript. In accordance with your suggestion, we have added a hypothesis section and reduced the redundancy in the introduction section.

- in the material and method, the soil used is presented only in terms of texture (line 100-101), additional information on pH, organic matter content, etc., may be useful.

Thank you for your recommendation. Unfortunately, we do not have pH data for all generations. We were unable to collect soil nutrient data for the experiment.

- Bulk soil samples (line 122) ...what size did you use?

Thank you for your comment. If the reviewer's query is referring to the size of the sieve, it is 3 mm mentioned in [lines 96-97]. If the query refers to the bulk soil inoculum: sterile potting soil ratio, we have added additional details in the methods section of the updated manuscript to address this query [lines 124 to 126]. Your input is highly appreciated. 

Round 2

Reviewer 1 Report

Comments and Suggestions for Authors

The insightful additional comments have significantly contributed to the improvement of the manuscript. Taking into account the current state of the paper, I proceed with its acceptance. However, to enhance the overall quality of the published paper, I have a few comments and suggestions to further refine the paper and ensure its excellence upon publication.

    1. - Please evaluate and ensure the effectiveness and quality of any figures or tables included. Verify that they are clear.

    2. - Double-check and perform minor editing of the English language to ensure clarity and precision. This will enhance the overall readability and professionalism of the manuscript.

  1.  
  1.  
  2.  

Comments on the Quality of English Language

Minor editing of English language required

Author Response

Dear Reviewer 2,

Thank you for accepting our publication and for dedicating your valuable time to reviewing this manuscript. Your suggestions are greatly appreciated, and we have addressed them below.

- Please evaluate and ensure the effectiveness and quality of any figures or tables included. Verify that they are clear.

Response: Thank you for your valuable time in reviewing this manuscript. Following your suggestion, we have uploaded high-resolution (300 dpi) TIFF files for all figures.

- Double-check and perform minor editing of the English language to ensure clarity and precision. This will enhance the overall readability and professionalism of the manuscript.

Response: Thank you for your valuable recommendation. Following your suggestion, we have meticulously incorporated phrasing changes and revised the titles of the results section to enhance comprehension. If there are specific paragraphs that you would like us to update, please do let us know. Your feedback is greatly appreciated.
